# Partial transfusion: on the expressive influence of trainable batch norm parameters for transfer learning

**Fahdi Kanavati**[1]                                          FKANAVATI@MEDMAIN.COM

**Masayuki Tsuneki**[1]                                        TSUNEKI@MEDMAIN.COM

[1]*Medmain Research, Medmain Inc., 2-4-5-104, Akasaka, Chuo-ku, Fukuoka, 810-0042 Japan*

## Abstract

Transfer learning from ImageNet is the go-to approach when applying deep learning to medical images. The approach is either to fine-tune a pre-trained model or use it as a feature extractor. Most modern architecture contain batch normalisation layers, and fine-tuning a model with such layers requires taking a few precautions as they consist of trainable and non-trainable weights and have two operating modes: training and inference. Attention is primarily given to the non-trainable weights used during inference, as they are the primary source of unexpected behaviour or degradation in performance during transfer learning. It is typically recommended to fine-tune the model with the batch normalisation layers kept in inference mode during both training and inference. In this paper, we pay closer attention instead to the trainable weights of the batch normalisation layers, and we explore their expressive influence in the context of transfer learning. We find that only fine-tuning the trainable weights (scale and centre) of the batch normalisation layers leads to similar performance as to fine-tuning all of the weights, with the added benefit of faster convergence. We demonstrate this on a variety of seven publicly available medical imaging datasets, using four different model architectures.

**Keywords:** transfer learning, batch normalisation, deep learning, medical imaging

## 1. Introduction

Transfer learning has been used for many medical imaging applications (Esteva et al., 2017; Wang et al., 2017; Menegola et al., 2017; De Fauw et al., 2018; Irvin et al., 2019). Typically, there are two approaches for doing it: (1) by using the pre-trained model as a feature extractor, followed by training a classifier with those features (Sharif Razavian et al., 2014), and (2) by fine-tuning the pre-trained model. The latter approach tends to lead to better performance (Litjens et al., 2017). Fine-tuning typically consists of tuning all of the layers (Girshick et al., 2014), or only a subset of the top layers (Long et al., 2015), with a low learning rate to avoid destroying the pre-trained weights. Other approaches have looked into learning which layers to tune based on the input (Guo et al., 2019).

While transfer learning is a commonly used approach, there is still uncertainty about whether transfer learning from ImageNet confers any advantages in performance compared to training from scratch, given the considerable differences in appearance between natural and medical images. Recent evidence seems to suggest that there is little benefit gained in evaluation performance from applying transfer learning to medical images, and simple, lightweight models trained from scratch were observed to have comparable performance to

larger fine-tuned models on chest x-ray and diabetic retinopathy datasets (Raghu et al., 2019). In addition, Raghu et al. (2019) observed that ImageNet performance was not predictive of performance on those datasets. Nonetheless, one of the primary advantages of transfer learning is faster convergence compared to training a model from scratch, especially when the aim is to use one of the latest convolutional neural network (CNN) architectures.

Within the context of fine-tuning a model, the batch normalisation (batch norm) (Ioffe and Szegedy, 2015) layer requires taking a few precautions due to how it operates differently between training and inference. During training, the layer uses the current batch mean and standard deviation to normalise the activations, and, at the same time, it updates exponentially moving averages of the mean and standard deviation and stores them as non-trainable weights to use during inference. While fine-tuning a model, it is usually recommended to use the batch norm layer in inference mode to avoid unexpected or poor performance on the validation and test sets from additional updates during training. Besides the moving average statistics, the batch norm layer has trainable weights representing affine parameters: scale $\gamma$ and offset $\beta$. Although these parameters are rarely investigated in isolation, there is recent evidence to suggest that they have high expressive power. Frankle et al. (2020) conducted experiments where only batch norm parameters were trained in a deep RestNet while the rest of the weights were randomly initialised and fixed. Despite the restrictiveness, this led to surprisingly high performance on CIFAR-10 and ImageNet, highlighting the expressive power of simply offsetting and scaling random features of a given architecture. Batch norm has also previously been investigated within the context of domain adaption. Li et al. (2018) proposed modulating the batch norm statistics from the source domain to the target domain for improved performance, and Chang et al. (2019) proposed training domain-specific batch norm layers while sharing all the other CNN parameters.

In this paper, we investigate the effect of the affine parameters of the batch norm layers within the context of transfer learning for medical images. To this effect, we compare a total of five different methods – four for transfer learning and one with random features – on nine datasets originating from a variety of seven publicly available datasets of medical images, using four different model architectures[1]. We find that (1) simply fine-tuning the batch norm affine parameters leads to similar performance as to fine-tuning all the model parameters, especially with DenseNet121; (2) fine-tuning leads to better performance than using the model as a feature extractor; and (3) using random weights and only training the batch norm parameters leads to acceptable performance on some datasets.

## 2. Datasets

We performed experiments using nine datasets that originated from the following seven publicly available datasets:

(a) Chest X-ray 17(Kermany et al., 2018) dataset consists of 5,856 180x180x3 px chest x-ray images from children labelled as having pneumonia or normal. The dataset was split into 4914 training, 320 validation, and 624 test. We used the original reserved test set.

---

1. Code available at https://github.com/fk128/batchnorm-transferlearning

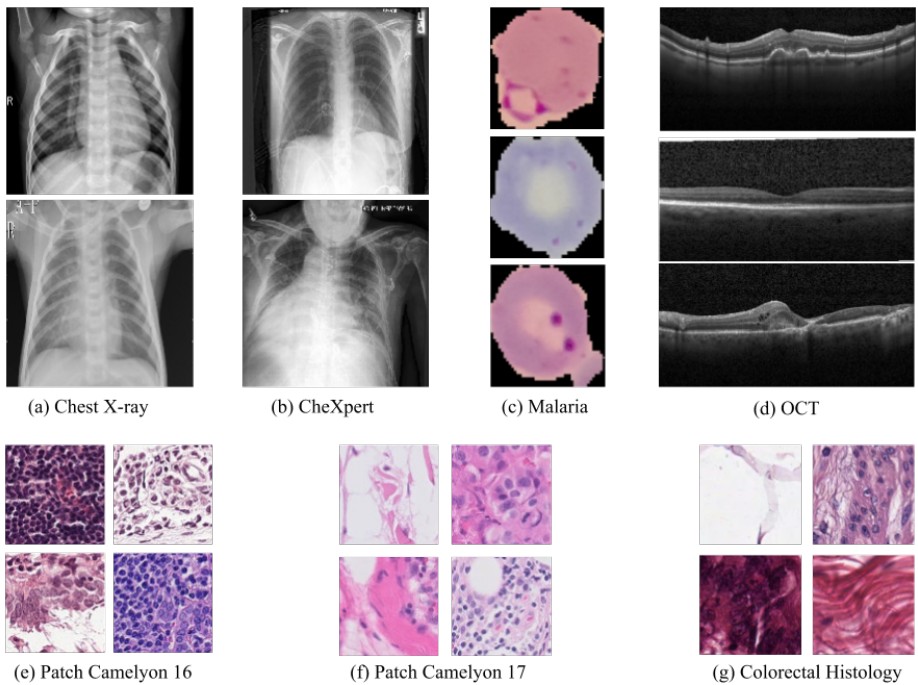

(a) Chest X-ray    (b) CheXpert    (c) Malaria    (d) OCT

(e) Patch Camelyon 16    (f) Patch Camelyon 17    (g) Colorectal Histology

Figure 1: Example images from each of the seven datasets.

(b) CheXpert (Irvin et al., 2019) dataset consists of 240,000 frontal and lateral chest x-ray images. Each image is labelled with up to 14 different thoracic diseases, with each multi-output label taking the values present, absent, or uncertain. We did not use the entire dataset, and we resized the images to 224x224x3 px and restricted to classifying only five pathologies: atelectasis, cardiomegaly, consolidation, edema, and pleural effusion, similarly to Raghu et al. (2019). In addition, we used the frontal images only, eliminated the ones with uncertain labels, and created subsampled, equally-balanced sets consisting of 11,633 training, 1,036 validation, and 1,350 test, ensuring that there was no patient overlap between the sets.

(c) Malaria dataset (Rajaraman et al., 2018) contains a total of 27,558 cell images with equal instances of parasitised and uninfected cells from the thin blood smear slide images of segmented cells. We resized the images to 120x120x3 px, and we randomly split the dataset into 70% training, 15% validation, and 15% test.

(d) OCT dataset (Kermany et al., 2018) consists of 108,309 OCT images with four categorical labels (choroidal neovascularization, diabetic macular edema, drusen, and normal). We subsampled this dataset into a smaller version and only used 3,200 images for training, 480 for validation, and the original equally-balanced test set of 1,000 images. We resized the images to 165x342x3px.

(e) Patch Camelyon 16 (Veeling et al., 2018) dataset consists of 327,680 96x96x3 px images extracted from whole-slide images of lymph node sections. Each image is annotated

| | DenseNet121 | ResNet50V2 | InceptionV3 | EfficientNetB3 |
|---|---|---|---|---|
| # of batch norm parameters | 83,648 | 45,440 | 17,216 | 87,296 |

Table 1: Total number of trainable batch norm parameters in each of the four model architectures.

with a binary label indicating presence of metastatic tissue. We used the original splits of 262,144 training, 32,768 validation, and 32,768 test. In addition to the full dataset, we used a smaller training version consisting of 6,400 training and 960 validation.

(f) Patch Camelyon 17 dataset is a patch-based variant of the Camelyon 17 challenge (Bandi et al., 2018) and prepared as part of the WILDS benchmark seeking to evaluate in-the-wild distribution shifts spanning diverse data modalities and applications (Koh et al., 2020). The dataset consists of 450,000 96x96x3px patches extracted from whole-slide images of breast cancer metastases in lymph node sections from five different hospitals. The patches can be grouped by hospital; therefore, we used the largest subset from one hospital as training (n=146,722), half of the patches of another hospital as validation (n=17,452), and the three remaining subsets as separate test sets (n=129,838, 85,054, and 59,436). In addition to the full dataset, we used a smaller training version consisting of 6,400 training and 960 validation.

(g) The Colorectal Histology (Kather et al., 2016) dataset consists of 5,000 150x150x3 px images each belonging to one of 8 classes. We randomly split the dataset into 70% training, 15% validation, and 15% test.

## 3. Method

### 3.1. Models

To perform transfer learning with a given pre-trained model, we removed the final classification layer, applied global average pooling, if not already applied, and then followed it with a dropout layer ($p = 0.5$), and finally applied a fully-connected classification (FC) layer with a number of outputs based on the given dataset. We used softmax activation if the dataset had categorical labels; otherwise, sigmoid. Figure E6 shows a diagram of the resulting model.

Most modern architectures contain a given number of batch norm layers. Table 3.1 lists the number of trainable batch norm parameters in each of the four model architectures that we have used: DenseNet121 (Huang et al., 2017), ResNet50V2 (He et al., 2016), InceptionV3 (Szegedy et al., 2016), and EfficientNetB3 (Tan and Le, 2019).

### 3.2. Batch normalisation layer

Given a batch of data with features $\mathbf{x}$, the batch norm layer computes the following as output

| | FC | FC-then-full | FC-then-BN | FC-BMA | FC-BN-RND |
|---|---|---|---|---|---|
| BN trainable parameters | no | yes | yes | no | yes |
| BN moving averages | no | no | no | yes | no |
| All other weights | no | yes | no | no | no (random) |
| FC | yes | yes | yes | yes | yes |

Table 2: Summary of which parameters were made trainable in each of the five methods.

$$\gamma \left( \frac{\mathbf{x} - \boldsymbol{\mu}}{\boldsymbol{\sigma} + \epsilon} \right) + \boldsymbol{\beta}. \tag{1}$$

The scale $\boldsymbol{\gamma}$ and offset $\boldsymbol{\beta}$ are the two trainable affine parameters, while the mean $\boldsymbol{\mu}$ and standard deviation $\boldsymbol{\sigma}$ are estimated based on the data. When the layer is in training mode, $\boldsymbol{\mu}$ and $\boldsymbol{\sigma}$ are computed based on the current batch, and, at the same time, their exponentially moving averages are updated and stored as non-trainable weights. In inference mode, the stored moving averages are used. This dual use mode is the reason why precautions are necessary when fine-tuning the model. Performing updates to the stored moving averages with a limited amount of data can lead to a large discrepancy between training and evaluation. A distinction is made for a batch norm layer between trainable and training. The former means that the affine parameters are trainable, while the latter refers to how $\mu$ and $\sigma$ are computed.

### 3.3. Training

In total we trained the models using five different methods: four variations of transfer learning and one baseline using random weights. The training methods were as follows, with a summary in Tab. 3.3:

1. **FC**: using the pre-trained model as a feature extractor and only training the fully-connected classification (FC) layer;

2. **FC-then-full**: training the FC layer first until no further improvement on the validation loss, followed by fine-tuning of all the weights (including the FC layer). During training, the batch norm layer was set to be trainable and in inference mode;

3. **FC-then-BN**: training the FC layer first for one epoch and then training the FC layer and all the affine parameters of the batch norm (BN) layers. During training, the batch norm layer was set to be trainable and in inference mode;

4. **FC-BMA**: only updating the batch norm moving averages (MA) and training the FC layer. During training, the batch norm layer was set to be non-trainable and in training mode.

5. **FC-BN-RND**: Randomly initialising all the weights and only training the affine parameters of all the batch norm layers and the FC layer. During training, the batch norm layer was set to be trainable and in inference mode.

In all our experiments, we used an initial learning rate of 1e-3, except for when fine-tuning the full model where we used 1e-5 to avoid destroying the pre-trained weights after having trained the FC layer with a learning rate of 1e-3. During training, we applied step decay to the learning rate where it decayed by a factor of five if the validation loss plateaued after an epoch. Training stopped automatically when the validation loss had not improved after six epochs. The model with the lowest validation loss was used to evaluate on the test set. We used the Adam optimiser (Kingma and Ba, 2014) with $\beta_1 = 0.9$ and $\beta_2 = 0.999$. We used a batch size of 32 for all datasets, except Camelyon where we used 64.

We applied minimal data augmentation consisting of random translations up to 10% and random zoom up to 10%. We applied random flipping horizontally and vertical for all datasets except CheXpert, Chest X-ray, and OCT where we only applied horizontal random flipping.

## 4. Results

We performed three repetitions of each combination of method, model, and dataset, and we summarised the results in Figure 2, where we reported the receiver operator characteristic (ROC) area under the curve (AUC) computed on the test sets. For the datasets where the outputs were categorical or had more than one test subset, we computed the AUC per label/subset and then averaged all of the AUCs. Tables A3, A4, A5, and A6 in the Appendix provide further breakdown of the performances on the test sets with more than one label for DenseNet121.

We see in Figure 2 that fine-tuning only the batch norm parameters (FC-then-BN) led to similar performance as to fine-tuning all of the model parameters (FC-then-full) in almost all of the experiments. The exceptions where it slightly under-performed were for InceptionV3 on CheXpert, Patch Camelyon 16 small, and Patch Camelyon 17 small; and ResNet50V2 on CheXpert. InceptionV3 has the lowest number of trainable batch norm parameters; nonetheless, comparable performance was still observed on some datasets.

In addition, we see a confirmation of the following: (1) fine-tuning (FC-then-full and FC-then-BN) results in improved performance compared to only using the pre-trained model as a feature extractor (FC), and (2) updating the moving averages (FC-BMA) tends to result in a degradation of performance compared to keeping them fixed while using the model as a feature extractor (Appendix D shows further results from setting the batch norm layer in training mode during fine-tuning).

And finally, we see that using random weights (RND-FC-BN) and only training the batch norm parameters results mostly in the lowest performance compared to the other methods; nonetheless, acceptable performance was observed on some datasets such as Chest X-ray and Malaria with DenseNet121; this is potentially due to the them being easier to classify. The EfficientNetB3 model with random weights was unable to train properly and the outputs remained saturated preventing the gradients from flowing. Setting the batch norm layer in training mode allowed the EfficientNetB3 model to train on some datasets (see Appendix B); however, it still had the lowest performance amongst the four models. This is in contrast to DenseNet121 and ResNet50V2 which contain shortcut connections that facilitate the flow of gradients during training with random weights.

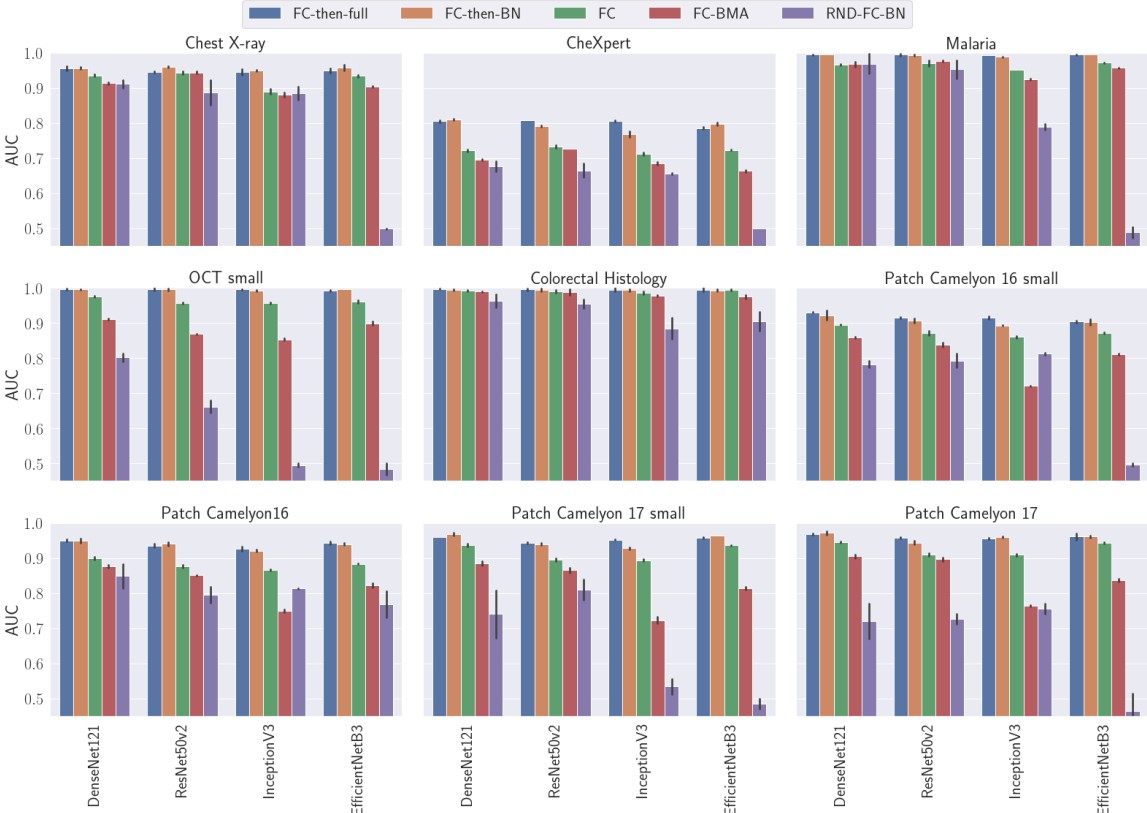

Figure 2: Barplots with standard deviation bars for the ROC AUC for the combinations of five different methods, nine datasets, and four model architectures. We see that training only the batch norm affine parameters (FC-then-BN) results in similar performance as to training the full model parameters (FC-then-full) in almost all of the experiments.

## 5. Discussion

We conducted experiments on a variety of different medical imaging datasets of various sizes ranging from x-ray, OCT, and histopathology, and our results demonstrate that fine-tuning only the batch norm affine parameters leads to similar performance as to fine-tuning all of the model parameters. We find this to be an interesting observation. Simply fine-tuning the batch norm affine parameters leads to faster convergence as there are fewer parameters to train and a higher learning rate can be used. Either training the fully-connected layer first followed by fine-tuning the batch norm layers, or training both simultaneously results overall in similar performance (see Appendix C).

A limitation of this study is that we did not perform any hyperparameter search optimisation apart from adopting settings used in common practice, given the large number of experiments already run. Another limitation is that we did not delve deeper into the primary mechanism as to why fine-tuning batch norm parameters is enough. Frankle et al. (2020) had observed using ResNets on CIFAR-10 and ImageNet that training the same number of randomly-selected parameters per channel performs far worse than training the batch norm parameters, and that the primary expressive power of batch norm feature comes from the ability to sparsify features. Our results clearly indicate that pre-trained convolutional layers are better than layers with randomly initialised weights, at least within the context of transfer learning. However, far larger random networks, especially in width, could potentially match the performance of pre-trained networks, given that wider ResNets with random weights exhibited improved performance on ImageNet (Frankle et al., 2020).

## 6. Conclusion

Our results demonstrate that fine-tuning only the batch norm affine parameters leads to similar performance as to fine-tuning all of the model parameters on a variety of medical imaging datasets. This overall results in faster convergence from the use of a higher learning rate and the fine-tuning of a smaller number of parameters without loss in performance. We observed this result with four different model architectures, and in particular with DenseNet121, highlighting the expressive power of simply scaling and offsetting outputs of pre-trained convolutional layers for transferring to new tasks.

### Acknowledgments

We are grateful for the support provided by Michael Rambeau, Meng Li, Kengo Tateishi, and Osamu Iizuka at Medmain Inc.

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

## Appendix A. Performances on test sets using DenseNet121

| Dataset | Method | Hosp. 0 | Hosp. 2 | Hosp. 3 |
|---|---|---|---|---|
| Patch Camelyon 17 | FC | 95.0 (0.1) | 93.2 (0.2) | 95.5 (0.2) |
| | FC-BMA | 84.6 (0.4) | 93.1 (0.1) | 94.5 (0.4) |
| | FC-then-BN | 96.5 (1.3) | 97.9 (0.3) | 97.2 (0.7) |
| | FC-then-full | 96.2 (0.8) | 96.9 (0.1) | 97.4 (0.0) |
| | RND-FC-BN | 78.3 (3.1) | 71.7 (8.8) | 65.6 (8.3) |
| Patch Camelyon 17 small | FC | 94.0 (0.7) | 92.7 (0.5) | 94.5 (0.7) |
| | FC-BMA | 83.1 (0.2) | 91.3 (0.5) | 92.6 (0.3) |
| | FC-then-BN | 97.0 (1.0) | 96.1 (0.4) | 97.3 (0.2) |
| | FC-then-full | 96.8 (0.2) | 95.0 (0.2) | 96.4 (0.1) |
| | RND-FC-BN | 64.3 (16.6) | 79.6 (3.2) | 79.3 (7.3) |

Table A3: Patch Camelyon 17 ROC AUC (mean and std in %) results for the three test sets each, originating from a different hospital.

| Method | Atelectasis | Cardiomegaly | Consolidation | Edema | Pleural Effusion |
|---|---|---|---|---|---|
| FC | 64.6 (0.7) | 69.7 (0.6) | 69.9 (0.3) | 77.7 (1.0) | 78.6 (0.4) |
| FC-BMA | 60.3 (0.2) | 69.1 (0.5) | 66.3 (0.4) | 78.3 (0.4) | 74.3 (0.3) |
| FC-then-BN | 70.5 (0.3) | 83.2 (1.2) | 78.9 (0.1) | 84.4 (0.2) | 87.7 (0.4) |
| FC-then-full | 69.5 (0.3) | 83.8 (0.7) | 77.4 (0.9) | 84.8 (0.4) | 87.1 (0.8) |
| RND-FC-BN | 59.7 (1.1) | 64.3 (2.3) | 64.1 (2.6) | 74.1 (1.7) | 74.5 (1.9) |

Table A4: CheXpert ROC AUC (mean and std in %) results for each of the five labels.

| Method | CNV | DME | DRUSEN | NORMAL |
|---|---|---|---|---|
| FC | 95.2 (0.2) | 97.9 (0.2) | 97.8 (0.2) | 99.1 (0.2) |
| FC-BMA | 91.5 (0.6) | 96.4 (0.3) | 81.1 (0.1) | 96.4 (0.6) |
| FC-then-BN | 99.0 (0.4) | 99.9 (0.0) | 99.3 (0.4) | 100.0 (0.0) |
| FC-then-full | 99.1 (0.3) | 99.9 (0.0) | 99.5 (0.6) | 100.0 (0.0) |
| RND-FC-BN | 92.2 (2.1) | 75.1 (1.7) | 66.1 (2.3) | 86.8 (1.0) |

Table A5: OCT small ROC AUC (mean and std in %) results for each of the four labels.

| Method | Tumor | Stroma | Complex | Lympho |
|---|---|---|---|---|
| FC | 99.5 (0.3) | 98.3 (0.3) | 97.7 (0.6) | 99.6 (0.1) |
| FC-BMA | 99.6 (0.3) | 98.3 (0.2) | 96.0 (0.3) | 99.8 (0.2) |
| FC-then-BN | 99.7 (0.5) | 99.4 (0.1) | 98.5 (0.6) | 99.4 (0.5) |
| FC-then-full | 100.0 (0.0) | 99.3 (0.1) | 99.0 (0.3) | 99.9 (0.0) |
| RND-FC-BN | 96.0 (2.1) | 94.7 (0.8) | 94.9 (0.7) | 97.9 (2.2) |
| | Debris | Mucosa | Adipose | Empty |
| FC | 98.9 (0.2) | 99.8 (0.2) | 99.8 (0.5) | 99.9 (0.3) |
| FC-BMA | 99.2 (0.7) | 99.7 (0.3) | 100.0 (0.0) | 100.0 (0.0) |
| FC-then-BN | 99.3 (0.4) | 99.7 (0.2) | 99.9 (0.1) | 99.3 (0.8) |
| FC-then-full | 99.7 (0.0) | 100.0 (0.0) | 100.0 (0.0) | 100.0 (0.0) |
| RND-FC-BN | 94.1 (2.5) | 96.0 (2.2) | 99.1 (1.1) | 98.4 (2.3) |

Table A6: Colorectal Histology ROC AUC (mean and std in %) results for each of the eight labels.

## Appendix B. Random weights

The EfficientNetB3 model with random weights was unable to train the batch norm parameters on some datasets where the outputs remained mostly saturated, preventing the gradients from flowing properly. Better random initialisation of the weights could potentially lead to better performance; however, this is outside the scope of this paper. We performed additional experiments where we have set the batch norm layers in training mode, and this allowed the model to train. The additional standardisation of the activations potentially provided better conditions to reduce the amount of saturated activations. Figure B3 shows the AUC for DenseNet121 and EfficientNetB3 with random weights and the batch norm layers either set to trainable and inference mode (RND-FC-BN), or trainable and training mode (RND-FC-BN-BMA).

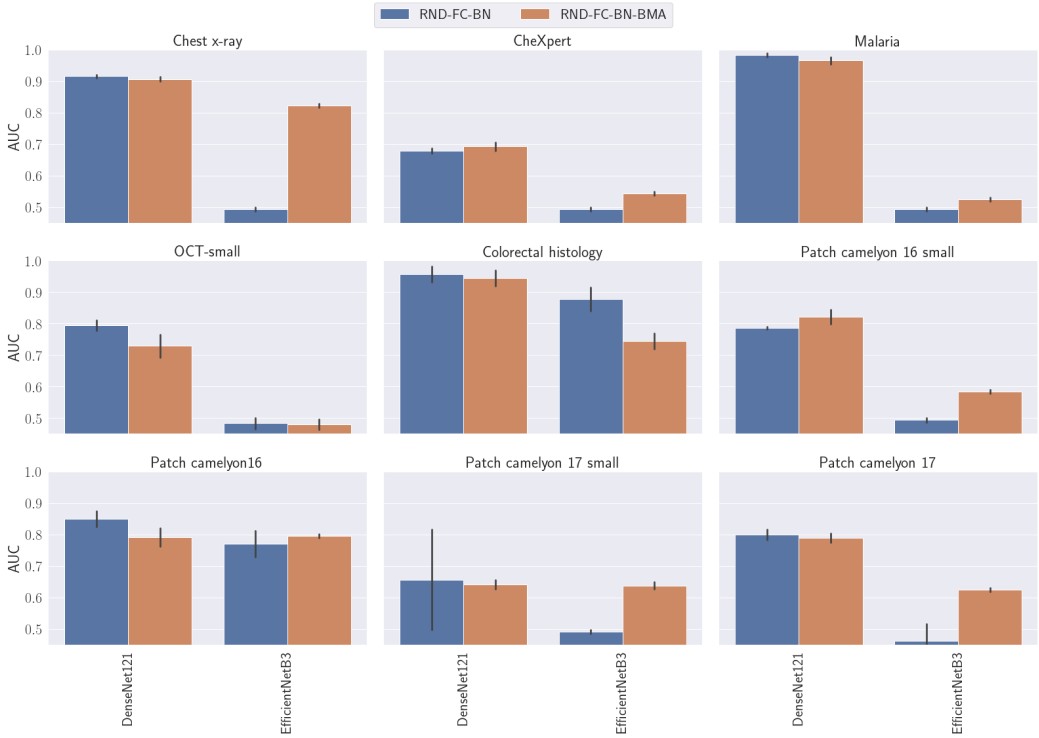

Figure B3: ROC AUC results for DenseNet121 and EfficientNetB3 with random weights with the batch norm layers either set to trainable and inference mode (RND-FC-BN),or trainable and training mode (RND-FC-BN-BMA).

## Appendix C. Training method with batch norm layer

Figure C4 shows ROC AUC results for the DenseNet121 and EfficientNetB3 models obtained using three different strategies for training the batch norm parameters and the FC layer.

1. **FC-BN** training the batch norm parameters and the FC layer together from the start with an initial learning rate of 1e-3.

2. **FC-then-BN** training the FC layer for one epoch, then training the batch norm parameters with the FC layers with an initial learning rate of 1e-3.

3. **FC-then-BN-lr** training the FC layer till no improvement on the validation set, then training both the FC and batch norm parameters with an initial learning rate of 1e-5.

Overall, training FC-BN and FC-then-BN exhibited similar performance. Lower performance was occasionally obtained with FC-then-BN-lr suggested there is some benefit in a higher learning rate.

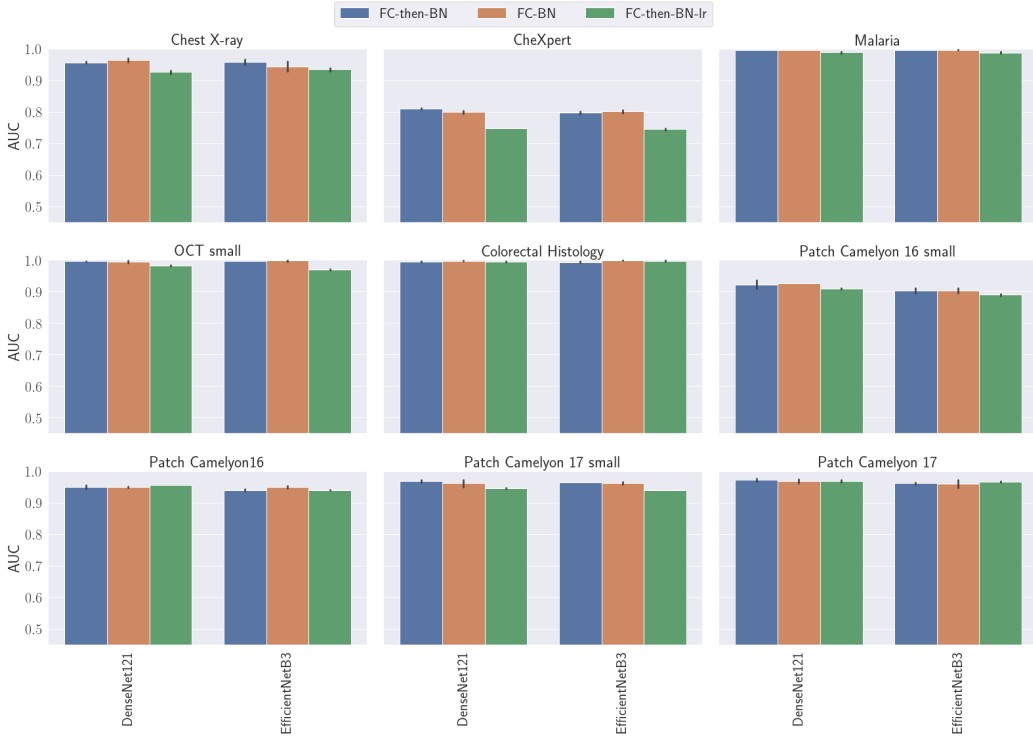

Figure C4: ROC AUC results for the DenseNet121 and EfficientNetB3 models obtained using three different strategies for training the batch norm parameters and the FC layer.

## Appendix D. Batch norm in training mode

Figure D5 shows the ROC AUC for three main transfer learning methods where we set for each the batch norm layers either in training or inference mode. Overall, setting the batch norm layers in inference model performs best for all methods.

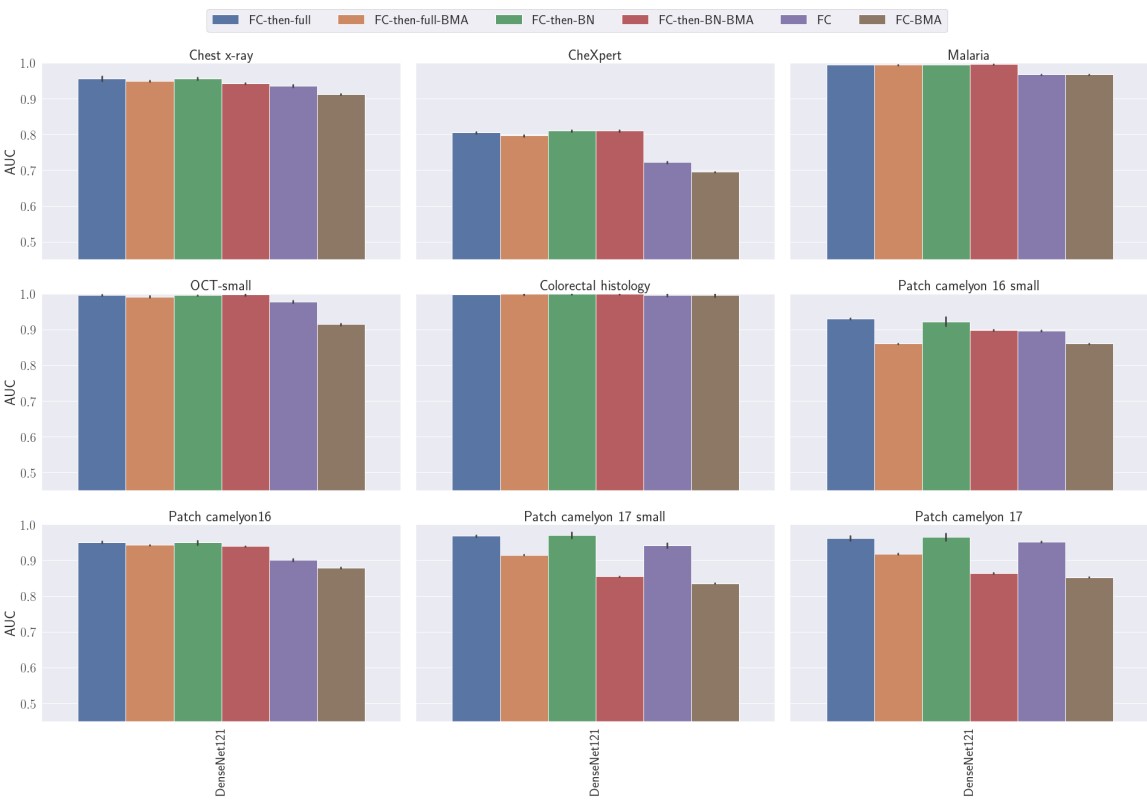

Figure D5: ROC AUC for three main transfer learning methods where we set for each the batch norm layers either in training (-BMA) or inference mode.

## Appendix E. Transfer learning model

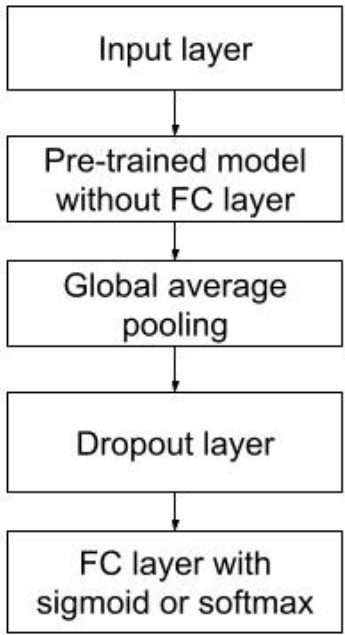

Figure E6: A plot of the transfer learning model

