# OpenReview forum: "Partial transfusion: on the expressive influence of trainable batch norm parameters for transfer learning"
_MIDL.io/2021/Conference — MIDL 2021_

### Official Review · AnonReviewer1 · 2021-03-04

**Confidence:** 4
**Preliminary Rating:** 2
**Final Rating:** 3

**Summary:**

The paper examines different ways to configure the batchnorm component in a transfer learning setup from models pretrained from ImageNet to medical datasets. Experiments are performed on four architectures and nine datasets. The main finding is that only fine-tuning the affine parameters of the batchnorm module, together with the fully-connected classifier, could achieve on par performance with fully-fine-tuning all model parameters. This could result in a more efficient transfer.


**Strengths:**

1. The research area is important: research on the role of batch normalization for transfer learning is important for medical imaging with potentially high impact. Improvements in transfer efficiency could help a wide range of tasks.
2. Consistent results across various datasets: authors undertake experiments on X-ray, OCT, and Histology with roughly consistent results to support the main findings.

**Weaknesses:**

The paper, in its current form, is more like a summary of experiments and lacks insight and analysis. I think the paper could be more thorough in analyzing the impact of batchnorm as well as how fine-tuning batchnorm alone is sufficient.

1. In Section 5, the authors mentioned sparsification could be the potential reason for its effectiveness but does not show any empirical evidence regarding how the sparsity changes during fine-tuning.
2. According to the empirical results, Inception-v3 is less suitable for partial transfusion. The authors claimed it has fewer batchnorm parameters. Does this change the conclusion of the paper? The need for considering the number of batchnorm parameters when deciding whether to use partial transfusion?
3. The model architectures in the experiments are rather deep. It is not clear whether the same conclusion holds for shallower models, e.g., ResNet-18.

**Deanonymize Review:**

no

**Detailed Comments:**

For notations, the authors use “FC-full” in Section 3.3 but “FC-then-full” in Section 4. It is better to stay consistent. Same for “FC-then-BN.”


**Final Rating Justification:**

Although I am still not convinced partial transfusion could be used in every scenario due to the limited experiments presented in this paper, I do think this paper has some novelty and adds to the knowledge of transfer learning.

**Justification Of The Preliminary Rating:**

Although the authors carried out a large number of experiments to demonstrate the benefits of partial transfusion, the paper lacks analysis regarding how it works. The experiments are focused on very deep models and it is unclear how it works on shallower models, such as ResNet-18. The paper hints at some relationship between the number of batchnorm parameters and the effectiveness of partial transfusion, but doesn't clearly state the recommended scenarios for applying partial transfusion.

**Paper Type:**

both

**Questions To Address In The Rebuttal:**

Please see section Weaknesses.

**Special Issue:**

no

---

> ### Author Response · Authors · 2021-03-15
> **Thanks for taking the time to review our paper and we appreciate your feedback**
>
> Below we address the weaknesses point by point.
>
> > Weaknesses:
> The paper, in its current form, is more like a summary of experiments and lacks insight and analysis. I think the paper could be more thorough in analyzing the impact of batchnorm as well as how fine-tuning batchnorm alone is sufficient.
>
>
> Though we did not delve deeper into the primary reason why fine-tuning batch norm is sufficient, the experiments performed still served to provide evidence for the hypothesis that fine-tuning only the batch norm affine parameters leads to similar performance as to fine-tuning all of the model parameters. So we did not simply summarise experiments without a corresponding hypothesis and conclusion.
> We have found this to be an interesting result in and of itself. We do agree that it would merit  further analysis; however, it would not be simply limited to medical image applications.
>
> > In Section 5, the authors mentioned sparsification could be the potential reason for its effectiveness but does not show any empirical evidence regarding how the sparsity changes during fine-tuning.
> According to the empirical results, Inception-v3 is less suitable for partial transfusion. The authors claimed it has fewer batchnorm parameters. Does this change the conclusion of the paper? The need for considering the number of batchnorm parameters when deciding whether to use partial transfusion?
>
> The performance of inception-v3 was not consistent across datasets; however, despite having the least number of batch norm layers, it still performed comparably to fine-tuning the entire model on some datasets.This does not change the conclusion of the paper.  Obviously the presence of batch norm layers is a requirement and this would not work for networks that do not have any batch norm layers. However, it would be possible to attempt to insert batch norm layers into pre-trained networks that do not have any. For instance one could take a VGG16 network and insert batch norm layers after all the convolutional layers and then only fine-tune the batch norm layers. One could potentially do similarly with Inception-v3 where additional batch norm layers are inserted. It would be interesting to perform such experiments.
>
>
>
> > The model architectures in the experiments are rather deep. It is not clear whether the same conclusion holds for shallower models, e.g., ResNet-18.
>
> It would be interesting to benchmark a larger variation of models. The ResNet-18 has about 11.5M parameters which is close to the number of parameters in EfficientNetB3, 12M. They might not have the same depth; however, it provides a reference point in terms of number of parameters.
>
> > For notations, the authors use “FC-full” in Section 3.3 but “FC-then-full” in Section 4. It is better to stay consistent. Same for “FC-then-BN.”
>
> We have changed the notation to make it more consistent.

---

> > ### Comment · AnonReviewer1 · 2021-03-18
> > **Thanks for the response**
> >
> > Although I am still not convinced partial transfusion could be used in every scenario due to the limited experiments presented in this paper, I do think this paper has some novelty and adds to the knowledge of transfer learning. I change my rating to *Weak Accept.*

---

### Official Review · AnonReviewer2 · 2021-03-05

**Confidence:** 3
**Preliminary Rating:** 3
**Recommendation:** Poster
**Final Rating:** 3

**Summary:**

In this paper, the authors address the problem of transfer learning in the context of medical images classification. More precisely, they investigate the fine-tuning of the batchnorm layers, following the work of Frankle et al. about the expressivity of batchnorm. Experimental results show that fine-tuning only the batchnorm layers leads to the same level of performance as fine-tuning the whole network. This observation was done on 7 different medical imaging datasets and 4 different CNN architectures.

**Strengths:**

The paper is well written and well structured. It is easy to read and understand.
In particular, the research problem and the claims are clearly stated, which is really appreciated.

The idea and results obtained are very interesting.
The experiments were run and validated on different datasets and with different CNN architectures, which is a really good point.

**Weaknesses:**

The main weakness I see is that there are some missing scenarios in the experimental part to fully support the claims.
The FC scenario having similar results as FC-full is a good indication that the claim "fine-tuning only the batchnorm affine parameters leads to similar performance as fine-tuning all the model parameters" could be valid. However, I feel that FC-full has some particularities like a 2-step fine-tuning (first the FC layer then the rest). I would be interested to see the performance of the model in other scenarios where we fine-tune also the feature extractor, for example a scenario with a simple fine-tuning strategy where all elements are made trainable and fine-tuned at the same time.

Apart from that, I only have minor propositions to improve the paper:
- in 3.1, a figure could help to visualize the different operations done on the initial network.
- in 3.3, a table summarizing the different parameters changed could help to differentiate the different scenarios.
- also in 3.3, I have difficulties understanding which elements are trained and which ones are frozen. Maybe the description of the scenarios could be extended to make this clearer.
- in the methodology section 3., I am missing a part where the different CNN architectures are listed and described. This information only appears first in the results section.


**Deanonymize Review:**

no

**Detailed Comments:**

In the last sentence of section 3.2, there is a small typo "batch norma layer" --> "batch norm layer"

**Final Rating Justification:**

The authors did a good work in the rebuttal and answered my questions/concerns. As mentioned by the other reviewers, the work presents some limitations in terms of generalization. However, the results could still be interesting for the community. For those reasons, I am maintaining my initial rating.

**Justification Of The Preliminary Rating:**

In general, the paper is of good quality. I enjoyed reading it and the results could be interesting for the community. I still have doubts about the claim, that I expressed in the weakness section, and I hope that the authors will be able to give some answers. The general presentation could also be improved. However, I think that the paper in the present state is already good for acceptance.

**Paper Type:**

both

**Questions To Address In The Rebuttal:**

Any answer to my previous feedback could be helpful.

**Special Issue:**

no

---

> ### Author Response · Authors · 2021-03-15
> **We really appreciate your time and feedback**
>
>
> > The main weakness I see is that there are some missing scenarios in the experimental part to fully support the claims. The FC scenario having similar results as FC-full is a good indication that the claim "fine-tuning only the batchnorm affine parameters leads to similar performance as fine-tuning all the model parameters" could be valid. However, I feel that FC-full has some particularities like a 2-step fine-tuning (first the FC layer then the rest). I would be interested to see the performance of the model in other scenarios where we fine-tune also the feature extractor, for example a scenario with a simple fine-tuning strategy where all elements are made trainable and fine-tuned at the same time.
>
>
> FC-full trains the fully connected layer first, but then the full model is trained, i.e. the feature extractor and the FC layer. So all the parameters are made trainable. The reason why we have performed the two step fine-tuning is because the FC layer starts with random weights, and starting by immediately training all the weights can result in a decrease in performance as the random weight might negatively affect the pre-trained weights. We have found this to be empirically the case and therefore simply used the common practice of 2-step fine-tuning approach (this is described here https://www.tensorflow.org/guide/keras/transfer_learning).
>
>
> > in 3.1, a figure could help to visualize the different operations done on the initial network.
>
> We have added supplementary Figure E6 to show the operations.
>
> > in 3.3, a table summarizing the different parameters changed could help to differentiate the different scenarios.
>
> We have added Table 2 to highlight which parameters were made trainable for each scenario.
>
>
>
> > also in 3.3, I have difficulties understanding which elements are trained and which ones are frozen. Maybe the description of the scenarios could be extended to make this clearer.
>
>
> We hope that the additional table 2 helps to make this clear.
>
> > in the methodology section 3., I am missing a part where the different CNN architectures are listed and described. This information only appears first in the results section.
>
> Given the page number limit, we felt it would be appropriate to simply defer it to the readers to consult the corresponding reference for each architecture.

---

### Official Review · AnonReviewer4 · 2021-03-08

**Confidence:** 5
**Preliminary Rating:** 3
**Recommendation:** Poster

**Summary:**

In this work authors compare different approaches of transfer learning, including fine-tuning the batch norm affine parameters, fine-tuning model parameters at different levels, for a variety of datasets. They in particular they show that only ﬁne-tuning the trainable weights (scale and centre) of the batch normalization layers leads to similar performance as to ﬁne-tuning all of the weights, with the added beneﬁt of faster convergence.

**Strengths:**

+ This work addresses a very relevant problem of domain adaptation and fine-tuning of deep learning in the field of medical imaging.
+ Authors use a variety of datasets for different relevant medical tasks to validate their results.
+ Methods use are simple but effective and could be utilized in general many medical imaging problems.

**Weaknesses:**

+ The list of experiments performed are very limited. For different datasets, 5 different training strategies were performed independently. As pointed by authors and in other prior works ( [Maithra Raghu, Chiyuan Zhang, Jon Kleinberg, and Samy Bengio. Transfusion: Understanding transfer learning for medical imaging. In Advances in neural information processing systems, pages 3347–3357, 2019](https://proceedings.neurips.cc/paper/2019/file/eb1e78328c46506b46a4ac4a1e378b91-Paper.pdf)), imagenet pre-training could be of only limited use. It would be more interesting to see impact of these approaches on sequential fine-tuning over these different datasets and tasks. For example, first the fine-tuning (starting from imagenet pre-trained weights) is performed on simpler medical tasks and then they are further fine-tuned on more complex tasks.
+ All the models used by authors have very large number of parameters. The reason for choice for such large models is not clear.
+ Authors only compare AUC for different datasets. As different datasets represents different medical tasks, AUC might not be as relevant for all of them.

**Deanonymize Review:**

no

**Justification Of The Preliminary Rating:**

This paper is a good demonstration of domain adaptation via transfer learning using simple techniques. However, the scope of shown work is little limited in terms of being generic. Given this is an application paper, I would expect a more detailed comparison of different strategies and comparisons with other similar methods in literature. If authors can add some of these, it could be a good poster presentation.

**Paper Type:**

validation/application paper

**Special Issue:**

no

---

> ### Author Response · Authors · 2021-03-15
> **Thanks for taking the time to review our paper, and we really appreciate your feedback.**
>
> Below we address the weaknesses point by point.
>
> > The list of experiments performed are very limited. For different datasets, 5 different training strategies were performed independently. As pointed by authors and in other prior works ( Maithra Raghu, Chiyuan Zhang, Jon Kleinberg, and Samy Bengio. Transfusion: Understanding transfer learning for medical imaging. In Advances in neural information processing systems, pages 3347–3357, 2019), imagenet pre-training could be of only limited use. It would be more interesting to see impact of these approaches on sequential fine-tuning over these different datasets and tasks. For example, first the fine-tuning (starting from imagenet pre-trained weights) is performed on simpler medical tasks and then they are further fine-tuned on more complex tasks.
>
> While pre-training could be of only limited use, the main advantage of it is much faster convergence compared to training from scratch. Fine-tuning a pre-trained model is still typically the first go-to method when training on a new dataset. In this paper the main hypothesis was that fine-tuning only the batch norm affine parameters leads to similar performance as to fine-tuning all of the model parameters. Therefore, the experiments we have performed compared fine-tuning the batch norm parameters with the typical methods of fine-tuning for transfer learning (e.g. https://www.tensorflow.org/guide/keras/transfer_learning).  To demonstrate that the hypothesis is true we have used 9 datasets, 4 different model architectures, and 5 variations of fine-tuning that attempt to isolate the effect of the batch norm layers, and we have repeated each experiment 3 times. We’re confident that constitutes enough evidence to indicate that the hypothesis is true.
>
> We do agree that it would be interesting to perform further variations, such as sequential fine-tuning, or spot fine-tuning as future work.
>
>
> > All the models used by authors have very large number of parameters. The reason for choice for such large models is not clear.
>
> As the intention is to perform transfer learning from pre-trained models, we have used models for which pretrained weights are easily available. The model sizes are in line of what has been used in prior publications applying transfer learning to medical images:
> DenseNet121 was used for the CheXpert [1] and as a baseline model for patch camelyon 17  domain shift benchmark [3]. Jeremy et al. [1] had also considered larger resnet models, ResNet152 and SEResNeXt101.
> InceptionV3 was used for retina and x-ray [2]
> ResNet50 and Inception-v3 were used for the study of transfer learning for medical images [4]
>
> This is why we have found it appropriate to use large models.
>
> [1] Irvin, Jeremy, et al. "Chexpert: A large chest radiograph dataset with uncertainty labels and expert comparison." Proceedings of the AAAI Conference on Artificial Intelligence. Vol. 33. No. 01. 2019.
> [2] Kermany, Daniel S., et al. "Identifying medical diagnoses and treatable diseases by image-based deep learning." Cell 172.5 (2018): 1122-1131.
> [3] Pang Wei Koh, et al. Wilds: A benchmark of in-the-wild distribution shifts. arXiv preprint arXiv:2012.07421, 2020.
> [4] Maithra Raghu, et. al. Transfusion: Understanding transfer learning for medical imaging. In Advances in neural information processing systems, pages 3347–3357, 2019
>
> > Authors only compare AUC for different datasets. As different datasets represents different medical tasks, AUC might not be as relevant for all of them.
>
> Sensitivity and specificity -- and the corresponding ROC AUC -- are typically relevant within the medical context. We are also using it as a comparative metric within each dataset and not across datasets. None of the test sets are highly imbalanced for which ROC AUC might be misleading and would merit looking at other metrics. For all the datasets that we have used, ROC AUC values have also been reported in their original corresponding papers.

---

### Meta-Review · Area_Chairs · 2021-03-25

**Recommendation:** Accept (Poster)

**Metareview:**

This paper introduces a comparison between transfer learning approaches, namely: fine-tuning the batch norm affine parameters, and fine-tuning model parameters at different levels.  Experiments were conducted on 9 datasets, 4 models, and 5 variations of fine-tuning that target the testing of the following hypothesis: fine-tuning only the batch norm affine parameters leads to similar performance as to fine-tuning all of the model parameters.  All reviewers acknowledged the value of the paper, saying that it is a valuable contribution to the community, even though it lacks a thorough analysis on the reason why fine-tuning batch norm leads to similar performance as to fine-tuning all of the model parameters.  I recommend the publication of the paper as poster.

**Paper Type:**

both

---

### Decision · Program_Chairs · 2021-03-31

Accept